# Selective Sulfur Dioxide Absorption from Simulated Flue Gas Using Various Aqueous Alkali Solutions in a Polypropylene Hollow Fiber Membrane Contactor: Removal Efficiency and Use of Sulfur Dioxide

**DOI:** 10.3390/ijerph18020597

**Published:** 2021-01-12

**Authors:** Hyun Sic Park, Dongwoan Kang, Jo Hong Kang, Kwanghwi Kim, Jaehyuk Kim, Hojun Song

**Affiliations:** 1Green Materials & Processes R&D Group, Korea Institute of Industrial Technology, 55 Jongga-ro, Jung-gu, Ulsan 44413, Korea; sic0722@kitech.re.kr (H.S.P.); dongwoan@kitech.re.kr (D.K.); jhkang7@kitech.re.kr (J.H.K.); kwanghwi@kitech.re.kr (K.K.); 2Department of Civil and Environmental Engineering, Pusan National University, 2 Busandaehak-ro, 63beon-gil, Geumjeong-gu, Pusan 46241, Korea; jaehyuk.kim@pusan.ac.kr

**Keywords:** desulfurization, hollow fiber membrane contactor (HFMC), polypropylene (PP), sulfur oxide (SO_2_), ammonium sulfate ((NH_4_)_2_SO_4_), once-through mode

## Abstract

Hollow fiber membrane contactors (HFMCs) provide a large specific surface area. Thus, their significantly reduced volume provides an advantage compared to the conventional gas–liquid contactor. In this study, the selective removal efficiency of flue gas, in which sulfur oxide (SO_2_) and carbon dioxide (CO_2_) coexist, was measured using a polypropylene (PP) HFMC with such advantages. To increase the selective removal efficiency of SO_2_, experiments were conducted using various alkaline absorbents. As a result, with 0.05 M ammonia solution, the removal efficiency of 95% or more was exhibited with continuous operation for 100 h or more. We confirmed that the absorbent saturated by the once-through mode was aqueous ammonium sulfate ((NH_4_)_2_SO_4_) solution and could be used as a fertilizer without additional processing.

## 1. Introduction

Despite the continuous development of new and renewable energies, fossil fuel consumption is not expected to decrease significantly worldwide. According to a report by the International Energy Agency (IEA), fossil fuels’ share in primary energy will drop below 50% in 2040 compared to 67% in 2017 [1,2]. Due to the COVID-19 worldwide pandemic, carbon dioxide emissions are temporarily reduced (coal and oil use decreased by 7% and 8%, respectively), but greenhouse gas emissions are still maintained at high levels [3]. Carbon dioxide (CO_2_), which is a cause of global warming from fossil fuel combustion, and sulfur oxides (SO_X_), generated by sulfur contained in fuels, cause acid rain and fine dust, so they are materials that must be treated [4]. Sulfur oxides are typically emitted from coal-fired power plants and ships due to the use of heavy fuel oil, which has a sulfur content of 3.5% [5]. The emitted sulfur oxides undergo photochemical reactions in the atmosphere to form fine particles several micrometers in diameter. Therefore, most countries regulate the concentration of sulfur oxides emitted from coal-fired power plants to less than 60 ppm [6,7,8]. For ships, the fuel’s sulfur content should be less than 0.5% according to 2020 International Maritime Organization (IMO) sulfur oxide emission regulations, which is a strengthened regulation compared to the existing 3.5% [9].

Methods of reducing sulfur oxides emitted from fossil fuel combustion are generally classified as either desulfurization during or after combustion [8]. During combustion, desulfurization is a method of collecting sulfur oxides generated in the combustion process in the form of CaS by introducing CaO [10]. Limestone is mainly used for desulfurization after combustion, and the method of collecting sulfur oxides with gypsum in a scrubber is widely used worldwide. Alternatively, an alkaline solution, such as Mg solution or seawater, is injected into a spray tower, packed column, or bubble column to selectively absorb and remove only SO_X_. However, in the process of producing by-product gypsum, a large amount of carbon dioxide is generated. Besides, energy consumption occurs during the drying process of gypsum dehydration, and, eventually, carbon dioxide is generated. There is an issue of depleting high-quality limestone in several countries, including Korea, such as CaO and CaCO_3_, required to produce by-product gypsum. Other scrubbers are faced with the challenge of increasing the efficiency of SO_X_ removal and reducing the volume of equipment by tightening environmental regulations [10,11,12,13]. Also, in the case of the secondary flue gas desulfurization (FGD) scrubber install to meet the SO_2_ emission standards, there are disadvantages such as channeling and flooding [14].

Membrane contactors are known to be efficient gas–liquid contact reactors that can significantly reduce the volume compared to conventional reactors because they can provide a large specific surface area (m^2^/m^3^) [15]. Besides, the investment cost and operation cost are lower than that of FGD scrubbers, so it is energy-efficient, it is easy to increase capacity by adjusting the number of modules, and it has the advantage of having excellent mass transfer properties [16]. According to a U.S. GTI Company report, when replacing the conventional FGD system with a membrane contactor, the total cost is reduced to 35%, the operating cost is reduced to 40%, and the footprint requirement is reduced to 40% [17]. Membrane contactors have been used to remove CO_2_ from exhaust gas from the viewpoint of mitigating global warming. The membrane contactor has excellent performance compared to other gas–liquid contactors because it flows into the hollow fiber membrane composed of polypropylene or the like. The solution can flow toward the outer shell and maintain a consistent interface between the solution and the gas through the membranes pores. Membrane contactors have disadvantages, such as pore wetting, which may occur because the solution continuously contacts the membrane [18,19,20,21]. However, these disadvantages can be solved by increasing the surface tension of the absorbent or increasing the hydrophobicity of the membranes surface; studies have been actively conducted to solve the problems.

Some papers have been published on selecting suitable absorbent for removing sulfur oxides using a membrane contactor [22,23,24,25,26,27,28,29]. In the literature, aqueous solutions, such as sodium hydroxide (NaOH) and sodium sulfite (Na_2_SO_3_), showed desirable absorption performance as SO_2_ absorbents [30,31]. However, these absorbents are uneconomical due to their high costs. In comparison, seawater, which has high SO_2_ removal efficiency, is economical. It has been reported as an alternative absorbent to conventional alkaline materials such as magnesium hydroxide and limestone. Still, various problems with using seawater have been reported, especially when seawater is discharged into the sea again after the SO_2_ removal process [26]. Hence, it is essential to select a suitable SO_2_ absorbent that efficiently removes SO_2_ removal and is economical and eco-friendly in membrane contactors.

In this study, various alkaline solutions were applied to a laboratory-scale membrane contactor composed of polypropylene (PP), a representative hollow fiber membrane material, to examine the selective removal efficiency of SO_2_ in the simulated flue gas of a massive oil boiler. We investigated the removal characteristics of SO_2_ according to the concentration and flow rate of each alkaline absorbent under constant SO_2_ composition and temperature conditions. The height of the transfer unit (HTU) was calculated using the results obtained from the SO_2_ removal experiment, and the performance was compared with the conventional separation process. The liquid by-products of experiments using ammonia solution were analyzed to determine the SO_2_ removal efficiency and the recyclability of recovered SO_2_, and the possibility of use as a liquid fertilizer is presented.

## 2. Materials and Methods

### 2.1. Materials

The commercial membrane contactor of the 3M Company used in this study is composed of hydrophobic polypropylene hollow fiber membranes (Liqui-Cel^TM^ MM-series, 3M Company, Charlotte, NC, USA). The specific surface area of the membrane contactor is 2000 m^2^/m^3^; the detailed specifications are summarized in Table 1.

Hydrogen peroxide solution (H_2_O_2_, 50 wt % in H_2_O, stabilized), sodium hydroxide (NaOH, ≥98%, anhydrous), sodium bicarbonate (NaHCO_3_, ≥99.7%), and ammonia solution (NH_4_OH, 28–30% for analysis, Supelco, EMSURE^®^, Merck KGaA, Darmstadt, Germany) were purchased from Sigma-Aldrich Co., Ltd. (Darmstadt, Germany), and sodium carbonate (Na_2_CO_3_, 99.5%, anhydrous) and sodium sulfate (Na_2_SO_3_, 96%, anhydrous) were purchased from DAEJUNG chemicals & metals (Siheung, Korea). Purchased the sea salt used to make the simulated seawater (Na^+^: 9300–9700 ppm; Cl^−^: 17,300–17,800 ppm; Mg^2+^: 1400–1450 ppm; Ca^2+^: 430–450 ppm: etc.) was from Qingdao Sea-Salt Aquarium Technology Co., Ltd. (Qingdao, China). Ultrapure distilled water (DI water) was obtained using a purification system (Human science Co., Ltd., Hanam, Korea). SO_2_ gas (1.00 cmoL/moL, N_2_ balance), CO_2_ gas (99.99%), and N_2_ gas (99.999%) were purchased from DEOKYANG Co., Ltd. (Ulsan, Korea).

### 2.2. Experimental Apparatus and Procedures

The membrane contactors facilities used in the experiment consisted of a mass flow controller (MFC), gas mixer, a gas analyzer (Sensoronic Co., Ltd., Bucheon, Korea), and membrane contactor. The setup is described in Figure 1.

The components of the simulated flue gas were used by mixing at concentrations of 2000 ppm for SO_2_ and 15% *v/v* for CO_2_ (N_2_ balance). The SO_2_ gas concentration was produced by referring to the concentration range (371–3500 ppm) used in typical FGD process studies [32,33]. The CO_2_ gas concentration was achieved by referring to the emission components of coal-fired power plants. The gas flow rate was fixed at 2 L/min, and the absorbent’s flow rate was changed to 25–150 mL/min to adjust the liquid to gas (L/G) ratio. The mixed gas and absorbents were applied to the countercurrent flow system in different directions, the mixed gas was injected to the shell side, and the absorbent was injected to the lumen side.

### 2.3. Mass Transfer of the Membrane Contactor

In the hollow fiber membrane contactors, the mass transfer between gas and liquid is achieved by a solution diffusion mechanism through a porous polymer membrane, as shown in Figure 2. The membrane contactor uses a porous hollow fiber membrane, unlike a general gas separation membrane that separates the gas phase under pressure conditions using a non-porous membrane. The membrane contactors used in this study contain thousands of microporous polypropylene hollow fibers knitted into an array wound around a center tube. This PP hollow fibers effective pore size is 0.04 micrometer, which has a porosity of 40%. The membrane is hydrophobic but has a gas-permeable surface, preventing absorbents from entering the pores, and the pores where gas exchange takes place are essentially filled with gas [34,35]. In Figure 2, P_g_, P_m_, and P_i_ represent the partial pressure at the interface between the gas, the membrane, and the gas–liquid phase, respectively. C_i_ and C_g_ represent the concentrations of the interface and gas, respectively [26,36]. K_l_ is the mass transfer coefficient in a liquid. If the reaction in which the gas is absorbed (or removed) by contact with the liquid occurs quickly or is an instantaneous reaction, the K_l_ value of the absorbents may be ignored. K_g_ is the mass transfer coefficient of gas and can explain the molecular diffusion between two gases in a mixed gas. Finally, K_m_ is the membranes mass transfer coefficients, determined by the membranes structural characteristics. (i.e., thickness, porosity, tortuosity of the pores, and diffusivity).

## 3. Results and Discussion

### 3.1. Characteristics of Hollow Fiber Membrane Contactor

Polymers, ceramics, and metals are used as materials for the membrane contactor. In metallic materials, the durability is the best, the swelling and wetting are low, and the chemical resistance is excellent. However, they are the costliest, and the processing and finely forming a thin film are difficult. Ceramic materials are also excellent in durability, chemical resistance, and thermal properties, but are fragile. For polymer materials, the membrane material is inexpensive compared to the previous two. It is possible to easily control the desired shape using a variety of polymers. However, there are disadvantages: swelling, wetting, and chemical resistance are weak [16,37]. For this reason, many studies considering polymer materials have used hydrophobic materials or increased hydrophobicity through a surface modification to minimize the wetting phenomenon [38,39].

In general, high hydrophobic polymers include polypropylene(PP), polyvinylidene fluoride(PVDF), and polytetrafluoroethylene(PTFE) [36,40]. PTFE (11.50 USD/m) and PVDF (0.36 USD/m), which have fluorine groups in the main chain, have high hydrophobicity and excellent physical and chemical properties. However, they are 36 and 1150 times more expensive than PP(0.01 USD/m), respectively [41]. Polymer membrane contactors are consumables and require regular replacement, so considering the replacement cost, using the relatively inexpensive PP, is desirable.

### 3.2. SO_2_ Removal Efficiencies of Various Absorbents

The absorption of SO_2_ gas was performed through a once-through mode. The removal rate was measured for one hour or until the SO_2_ removal rate was less than 50% using various absorbents (simulated seawater, H_2_O, H_2_O_2_, NaOH, NaHCO_3_, Na_2_CO_3_, Na_2_SO_3_, and NH_4_OH). The chemical equation for the SO_2_ removal reaction for the various absorbents used can be summarized as follows [26,33,42,43,44,45]:SO_2_ (g) ↔ SO_2_ (aq),(1)
H_2_O (aq) + SO_2_ (aq) ↔ H_2_SO_3_ (aq),(2)
H_2_O_2_ (aq) + SO_2_ (aq) ↔ H_2_SO_4_ (aq),(3)
2NaOH (aq) + SO_2_ (g) → Na_2_SO_3_ (aq) + H_2_O (aq),(4)
2NaHCO_3_ (aq) + SO_2_ (aq) → Na_2_SO_3_ (aq) + 2CO_2_ (g) +H_2_O (aq),(5)
Na_2_CO_3_ (s) + SO_2_ (aq) → Na_2_SO_3_ (aq) + CO_2_ (g),(6)
Na_2_SO_3_ (s) + SO_2_ (aq) + H_2_O (aq) → 2NaHSO_3_ (aq),(7)
2NH_4_OH (aq) + SO_2_ (aq) + H_2_O (aq) ↔ (NH_4_)_2_SO_3_ (aq) + 2H_2_O (aq),(8)

All absorbents except for simulated seawater and water were tested at a concentration of 0.01 M, which was relatively low compared to that used in previous studies. The performance of the absorbent was high, in the descending order of: NH_4_OH > H_2_O_2_ > Na_2_CO_3_ > NaHCO_3_ > NaOH > simulated seawater > Na_2_SO_3_ > DI water, as depicted in Figure 3. The same trend can be confirmed by looking at the SO_2_ loading of the absorbent over time in Figure 3.

When DI water was used as an absorbent, as reported in the results for general SO_2_ removal experiments, the removal rate rapidly decreased from the beginning of the experiment, and then showed a removal rate of 50% or less in less than 30 min [16,32,33,46,47]. The tendency for various types of absorbents in this study is not much different from those in the previous studies. However, the simulated seawater showed somewhat high SO_2_ removal efficiency for 30 min after the experiment.

In the hollow fiber membrane contactor, the mass transfer between gas and liquid is achieved by diffusion through the hollow fiber membrane pores.

The reaction region performs the absorption of gas through diffusion in the liquid phase. As the absorbent concentration decreases, the reaction region in the liquid phase that absorbs gas decreases, so that the gas absorption rate, that is, the SO_2_ removal efficiency, decreases. For this reason, we judged that the removal performance of simulated seawater was measured as being relatively high using an absorbent having a low concentration of 0.01 M. Besides, SO_2_ removal performance is closely related with the pH condition of the absorbent. According to Zhang et al., when the absorbent’s pH decreases, the mass transfer of SO_2_ from the gas phase to the absorbent decreases [48]. Therefore, the SO_2_ removal efficiency also decreases.

### 3.3. Removal Efficiency According to L/G and Concentration Changes of NH_4_OH

Figure 4 shows the SO_2_ removal efficiency according to the L/G change by the absorbent concentration selected through the previous experiment. In each experimental condition, the gas flow rate was fixed at 2 L/min, and the liquid flow rate was changed from 25 mL/min to 150 mL/min so that the L/G value was set from 12.5 to 75. The higher the concentration of the absorbent, the longer the reaction region in the absorbent mentioned above increased; thus, the duration of maintaining the SO_2_ removal efficiency of 90% or more lengthened. This is the same trend as reported in previous general research results. However, the L/G value used in this experiment is about four times lower than that in existing studies, which means that it is possible to process a larger amount of SO_2_ during the same time [26,33].

The absorbent concentration was tested in various conditions from 0.005 to 0.1 M, considering the housing material (polycarbonate) of the commercially available membrane contactor (3M Liqui-Cel^TM^ MM series). If the experiment is conducted with an absorbent concentration of 0.1 M or higher, the membrane contactor performance decreases due to corrosion and damage to the housing material. That is, the SO_2_ removal efficiency decreases. This problem can be overcome by replacing the membrane contactor housing material. It can be prevented by experimenting with the stability test with the absorbent material before the experiment.

### 3.4. Effects of Long-Term Operation

Based on the experiment results described above, the ammonia solution with a concentration of 0.05 M was selected as the optimal absorbent. The selected ammonia solution was operated for more than 100 h, with an L/G of 75 mL/L, and the results are shown in Figure 5. The lines marked with a black square symbol and a red circular symbol line indicate the SO_2_ and CO_2_ removal efficiencies, respectively. The transition point occurring in the graph indicates the replacement time of the used absorbent.

As a result, we confirmed that the removal efficiency of SO_2_ was 95% or more, but in CO_2_, the removal efficiency showed a negative value of 0% or less. The reason for this is that SO_2_ is far superior to CO_2_ in reactivity. Therefore, a certain amount of CO_2_ absorbed at the beginning of the reaction by the substitution reaction was removed, and the removal efficiency showed a negative value.

There was no problem with maintaining the SO_2_ removal efficiency above 95% for over 100 h in the long-term operation experiment. However, in the case of CO_2_, as the removal efficiency rapidly increased after 100 h, selective absorption of SO_2_ was not achieved. This is thought to be due to a decrease in material transfer performance due to corrosion of the membrane contactors potting material after 100 h.

When replacing the absorbent in long-term operation experiments, the aqueous mixture may contain ammonium carbonate (NH_4_)_2_CO_3_ absorbed a small amount of CO_2_. Ammonium carbonate is spontaneously decomposed into ammonium bicarbonate (NH_4_HCO_3_) and ammonia. At this time, ammonia is easily soluble in water, so there is no need to consider gaseous ammonia toxicity.

Ammonium carbonate ((NH_4_)_2_CO_3_) produced by CO_2_ and ammonia solutions reaction causes corrosion of polyurethane, the potting material of the membrane contactor used in the experiment [49]. The corrosion of the membrane contactor potting caused by ammonium carbonate did not occur suddenly, but rather slowly. The corrosion of the polyurethane gradually proceeded with each replacement cycle of the absorbent. An overflow of gas was observed in the potting of the membrane contactor after 100 h of operation. The corrosion phenomenon of potting of the membrane contactor results from confirmation through repeated experiments and data on the chemical stability of polyurethane.

Polyurethane and epoxy resin are commonly used as the potting material for hydrophobic hollow fiber membranes. Polyurethane has the advantage of being easy to work because it has a fast curing speed and is relatively soft. On the other hand, epoxy has a slow curing rate but chemically and temperature more resistant. Therefore, when using NH_4_OH as an absorbent under the condition of coexisting CO_2_, a material such as epoxy should be used to consider chemical stability as the membrane contactors potting material.

### 3.5. Overall Mass Transfer Coefficient for Various Absorbents

The overall mass transfer coefficient was calculated by inserting the SO_2_ removal efficiency results, the membrane contactors specification, and the gas flow rate into Equation (9). The calculated coefficient value indicates the performance of the absorbent for membrane contactors. The calculation equation is as follows [26]:K_G_a = (Q_g_/ZP) × log[(C_SO2_,_IN_)/(C_SO2_,_OUT_)],(9)
where K_G_a is the overall mass transfer capacity coefficient (kmol/m^3^·hr·kPa), Q_g_ is the gas flow rate, Z is the effective hollow fiber membrane length (m), and P is the operating pressure (kPa). In this study, the membrane contactors do not depend on the pressure during operation. The force applied to the membrane contactor is 1 atm, and the pressure difference is close to zero. It was confirmed that there is no difference when comparing the pressure applied to the membrane contactor through an experiment [33,50].

Figure 6 shows the overall mass transfer coefficients for SO_2_ of various absorbents over time. As the flow rate of the absorbent increases, the SO_2_ removal efficiency in the simulated flue gas increases, and the overall mass transfer coefficient increases continuously. These results indicated that the prepared absorbent has excellent SO_2_ material transfer capability. The calculated overall mass transfer coefficient was maintained almost constant when the absorbent flow rate was over 75 mL/min (L/G = 37.5), and the flow rate at this point was found to be the optimum condition.

The advantage of using K_G_a as the basis for comparison is that the effective contact area per unit volume describes the effective surface area in both the absorption tower and membrane contactor systems. The absorption towers and membrane contactor systems are beneficial because they have different packing densities, porosities, and total surface areas for mass transfer. However, it can approximate the membrane surface area with reasonable accuracy in the membrane contactor system, but the packed absorption towers wetting area is challenging to determine.

The separation process is based on the packing height (Z) value when calculating the theoretical process size [50]. The mass transfer height (HTU) used when calculating the Z-value is determined by the overall mass transfer coefficient and gas flow rate. Since the mass transfer height is used as a measure of general process efficiency, it can be confirmed by calculating the HTU value when calculating the ideal size and efficiency in the separation process. The formula is as follows:HTU = Q_g_/(K_G_a × S_MC_),(10)
where HTU is the mass transfer height, Q_g_ is the gas flow rate, S_MC_ is the cross-sectional area of the membrane contactor, and MC is membrane contactor.

Table 2 compares the HTU values of the conventional absorption tower and the set up to which the membrane contactor used in this study was applied. The HTU value calculated by applying the size of the membrane contactor module and the experimental value is 5 × 10^−2^; the volume is 1/36th when the membrane contactor is applied compared to the conventional packed column [26].

Packed columns, which are widely used, have high technology maturity. However, reported as the energy consumption of the conventional packed column is 0.58 kWh/Nm^3^-CO_2_, which can be reduced by about 33% when replaced by a membrane contactor system. Furthermore, the volume of absorption towers can be reduced by 63% with the application of the membrane contactor hybrid system compared to conventional gas absorption systems [52].

### 3.6. Reuse of By-Products

After the end of the experiment, we checked whether the absorbent reacts with SO_2_ by checking the absorbent pH value change. However, for more accurate structural analysis, Attenuated total reflectance-Fourier transform infrared spectroscopy (ATR-FTIR) spectrum analysis of the generated by-products was performed. Figure 7 compares the FTIR spectrum of 0.05 M aqueous ammonia solution used as an SO_2_ absorbent. (a) is an aqueous ammonia solution prepared at a concentration of 0.05 M; (b) is an absorbent when the SO_2_ removal efficiency was 99%; (c) is the IR spectrum result of the absorbent when the SO_2_ removal efficiency was reduced to 50% or less. The IR spectrum (d) of the ammonium sulfate solution was prepared at a concentration of 0.05 M and was used to confirm whether the prepared absorbent produced ammonium sulfate. As a result of IR analysis, we found that as the absorption of SO_2_ proceeds, the characteristic peak observed in the aqueous ammonium sulfate solution increased (1050–1150 cm^−1^, asymmetric stretch bend of SO_4_ ion; 1400–1450 cm^−1^, deformation or umbrella bend of NH_4_ ion), confirming that ammonium sulfate was formed [53].

In the ammonium sulfate solution produced by the once-through mode, 3 L of NH_4_OH (0.05 M) can theoretically produce about 7.43 g of ammonium sulfate. Ammonia nitrogen used as fertilizer means nitrogen in the form of NH_4_ or NH_3_ ions, and ammonia dissolved in water also exists as ammonia nitrogen used as fertilizer, so there is no need to remove it. However, it should be used following the appropriate concentration for each crop, and it is not suitable for acidic soils, so it should be used only in neutral or alkaline soils. Also, when mixed with basic fertilizer, ammonia volatilizes, so it should not be mixed.

The world demand and the global market of ammonium sulfate used as fertilizer is expected to reach 29.81 million tons and 3.44 billion dollars, respectively, by 2022 [54]. Ammonium sulfate fertilizer is generally suitable for crops such as common wheat, corn, rice, cotton, sweet potato, sesame seeds, fruit trees, and vegetables. The appropriate amount of fertilizer used depends on the crop. In the case of wheat, 62 kg of ammonium sulfate fertilizer per 0.1 ha is used. It is considered that a sufficient supply of ammonium sulfate aqueous solution is possible by increasing the absorbent concentration used in this study or by scale-up the reaction facility [55]. Conventional ammonium sulfate fertilizers are used in powder form at once, resulting in adverse effects such as pest damage and cold resistance reduction due to nitrogen excess. Since ammonium sulfate fertilizer is fast-acting, it is common to divide it several times. Additional water should be used to prevent denitrification after fertilization. However, ammonium sulfate produced through the experiment is a liquid form containing moisture, so it can be sprayed as is. It is expected to be suitable given the regulations and standards of use because it does not contain heavy metal. Also, the produced ammonium sulfate contains a large amount of water in the form of an aqueous solution. Therefore, if applied in practice, an aqueous ammonium sulfate solution should be transported to the tank lorry for use, or additional facilities (e.g., dryer, crystalizer) should be used in the production process to supply it in powder form [56,57]. Besides, ammonium sulfate fertilizer does not cause acidification, and the rice yield increases by about 3% in reclaimed lands, alpine regions, and limestone regions compared with urea fertilizers [58,59].

However, when real flue gas is used, it is necessary to install a pretreatment facility that removes trace amounts of pollutants that may absorb some harmful substances by the absorbent or reduce the membrane contactors performance. Pretreatment facilities are classified according to the type of pollutant. For example, cyclones, bag filters, and electrostatic separators (ESPs) are often used to remove contaminants from particulate pollutants. Among these, bag filters are widely used as technologies to minimize dioxin, VOCs, and heavy metals by combining activated carbon at the facility front-end [60]. Also, the flue gas temperature is 90–140 °C, and the operating temperature of the membrane contactors used is below 80 °C or less. When real flue gas is used, to replace it with membrane contactors applied to high temperatures, or operating temperatures will be adjusted through heat exchangers.

## 4. Conclusions

As a result of experimenting with the absorption of SO_2_ from the model of flue gas using various absorbents, we found excellent performances for the following in descending order: NH_4_OH, Na_2_CO_3_, NaOH, and H_2_O_2_. When the ammonia solution was used as an absorbent, it showed a high selective removal efficiency of SO_2_ gas. As the L/G ratio increased, mass transfer accelerated, and the removal efficiency was maintained for a long time. We concluded that an absorbent with a concentration of 0.05 M is suitable for maintaining the SO_2_ removal efficiency and understanding the replacement cycle during long-term operation.

When replacing the conventional packed column with a membrane contactor composed of PP, gas separation and the removal performance are maintained, and the volume of equipment is significantly reduced. The performance of conventional scrubbers and hollow fiber membrane contactors was compared in terms of increased contact surface area rather than the increased mass transfer coefficient. In the absorption process, the larger the interface area between the gas and the liquid, the higher its overall velocity affecting the absorption/removal performance. The membrane contactor has thousands of micro-sized hollow fiber membranes modulated by housing materials, providing a large contact area despite the small volume.

As a result of the experiment conducted with CO_2_ and SO_2_ coexisting, the SO_2_ removal performance decreased due to corrosion of the membrane contactors’ potting material. As a result, we confirmed that the wetting of the membrane contactor and the corrosion of the potting material decreased the separation and removal performance, thereby adversely affecting all processes of applying the membrane contactor. However, this problem can be overcome by replacing the potting material with a material with high chemical stability. Using a chemically stable potting material enables the application of a high-concentration absorbent to the membrane contactor.

Through this study, the possibility of the selective absorption of SO_2_ in flue gas and its reuse as a saturated absorbent fertilizer was confirmed using a once-through mode. If it can be applied at a higher concentration than the currently used absorbent, it is expected that long-term operation of the desulfurization facility will be possible, and the amount of ammonium sulfate, a by-product, will increase. Although many issues remain to be resolved, such as the concentration of SO_2_ emitted or the concentration of absorbents used, we hope that this technology will allow by-products of the absorption process to be reused as resources, not as waste.

## Figures and Tables

**Figure 1 ijerph-18-00597-f001:**
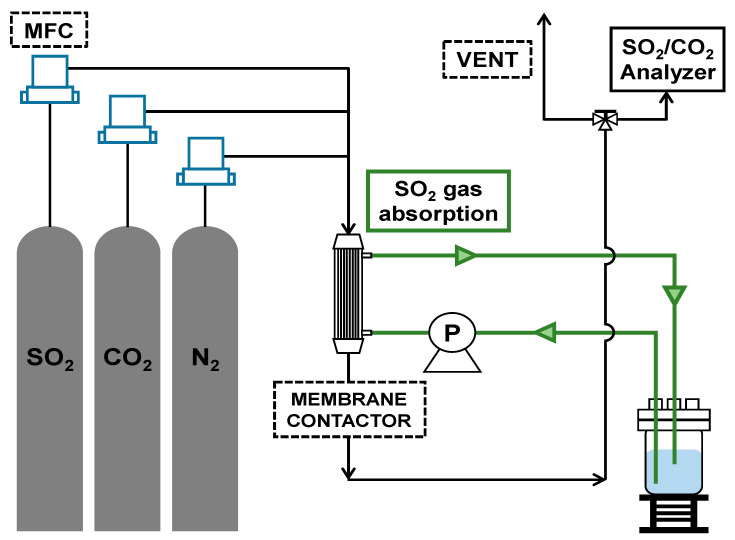
A schematic diagram of the experimental apparatus (MFC, mass flow controller).

**Figure 2 ijerph-18-00597-f002:**
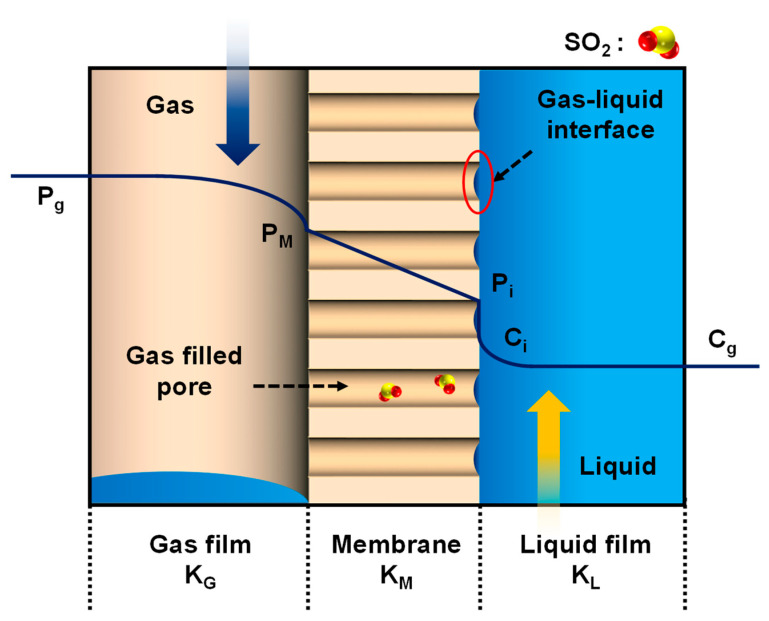
Mass transfer in hollow fiber membrane contactor. P_g_, P_m_, and P_i_ represent the partial pressure at the interface between the gas, the membrane, and the gas–liquid phase, respectively.

**Figure 3 ijerph-18-00597-f003:**
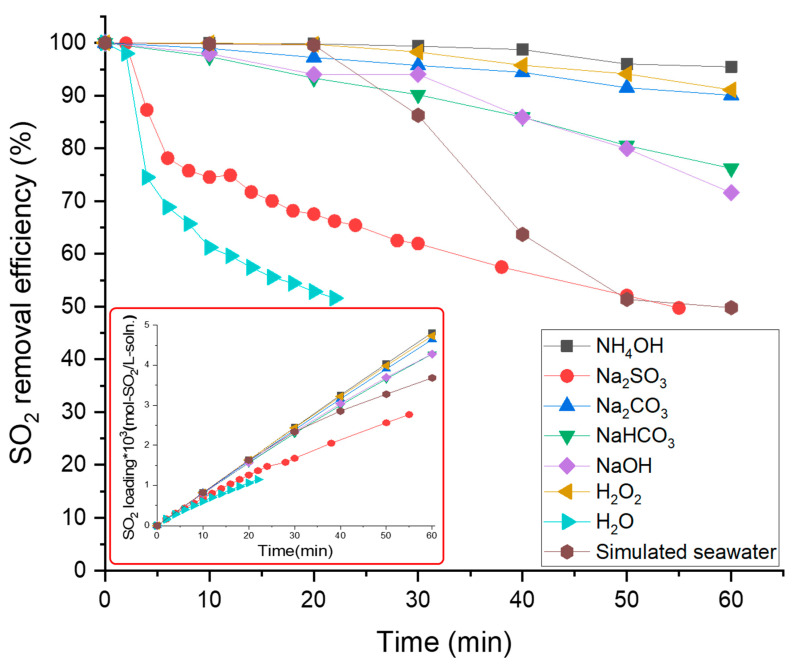
Effect of absorbents on SO_2_ gas removal efficiency and SO_2_ loading.

**Figure 4 ijerph-18-00597-f004:**
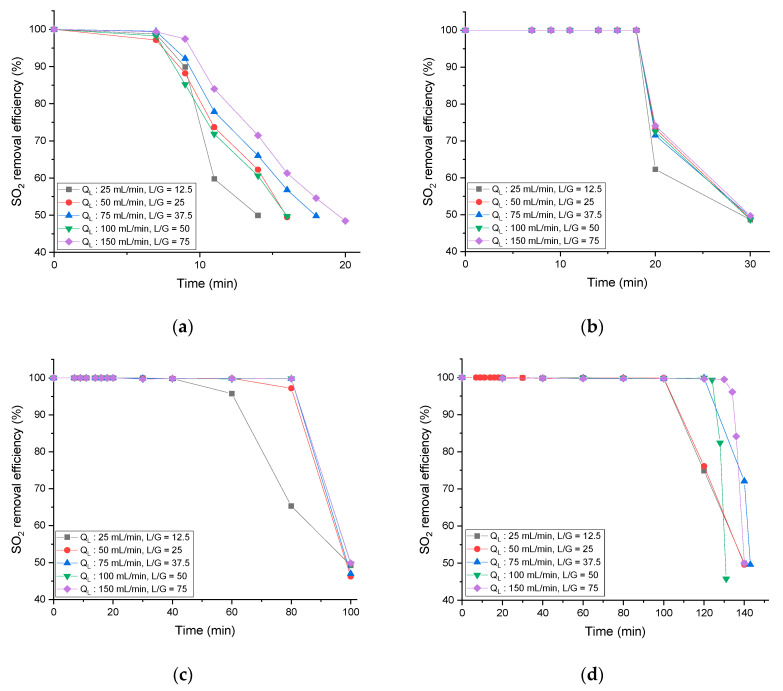
Effect of liquid flow rate and absorbent concentration on SO_2_ removal efficiency: (**a**) 0.005 M NH_4_OH; (**b**) 0.01 M NH_4_OH; (**c**) 0.05 M NH_4_OH; (**d**) 0.1 M NH_4_OH.

**Figure 5 ijerph-18-00597-f005:**
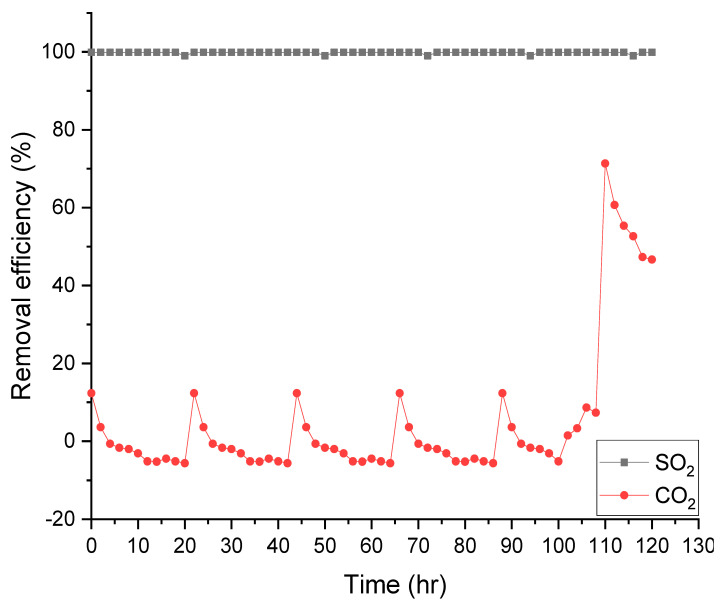
Long-term operation of membrane contactor (Absorbent concentrations: 0.05 M NH_4_OH, liquid to gas ratio (L/G) = 75 mL/L).

**Figure 6 ijerph-18-00597-f006:**
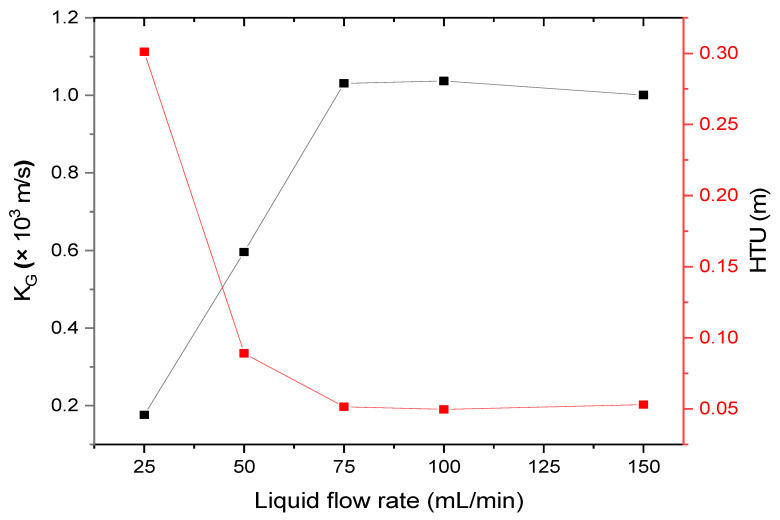
Effect of absorbents on SO_2_ overall mass transfer coefficient.

**Figure 7 ijerph-18-00597-f007:**
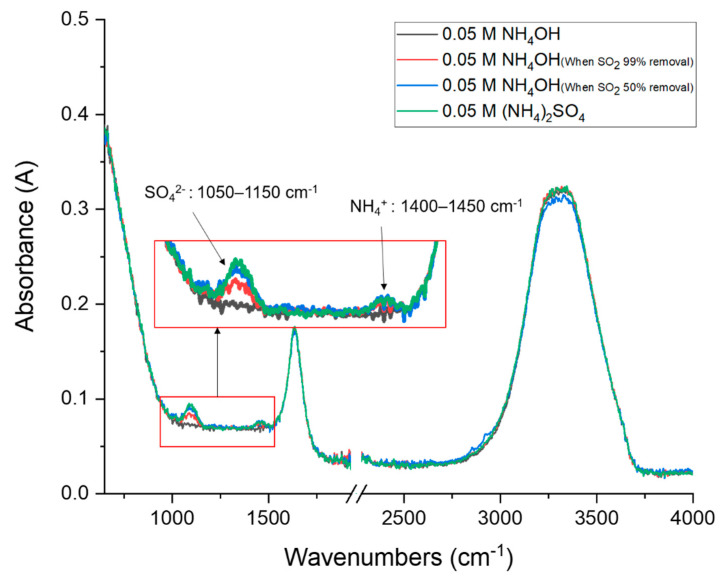
FTIR spectra of fresh and SO_2_ absorbed aqueous NH_4_OH.

**Table 1 ijerph-18-00597-t001:** Specifications of the hollow fiber membrane contactor module (3M Liqui-Cel™ MM-1 × 5.5 Series).

Parameter	Value
Module outer diameter (mm)	27
Module inner diameter (mm)	20
Module length (mm)	176
Fiber inner diameter (μm)	200
Fiber outer diameter (μm)	300
Fiber length (mm)	143
Effective pore size (μm)	0.04
Porosity (%)	40
Number of fibers	2000
Module volume (m^3^)	1.0 × 10^−4^
Contact area (m^2^)	0.2
Contact area/volume ratio (m^2^/m^3^)	2000

**Table 2 ijerph-18-00597-t002:** Comparison of mass transfer height (HTU) values with absorption towers of conventional desulfurization facilities.

	Packed Column [46]	Spray Tower [51]	Membrane Contactor [33]	Membrane Contactor (This Work)
Effective length of module (m)	1.07–3.21	1.75–4.13	0.21–2.32	0.15
Flow rate of gas (L/min)	2 L/min (3 × 10^−5^ m^3^/s)	-	-	-
SO_2_ removal efficiency (%)	>99	-	-	-
HTU (m)	0.6–1.8	0.98–2.32	0.12–1.3	0.05
Reactor volume	1	0.0061–0.0184	0.05–0.15	0.07–0.20

## Data Availability

The data presented in this study are openly available.

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
