# Peer review of "Selective Sulfur Dioxide Absorption from Simulated Flue Gas Using Various Aqueous Alkali Solutions in a Polypropylene Hollow Fiber Membrane Contactor: Removal Efficiency and Use of Sulfur Dioxide"

_ijerph, 2021, doi:10.3390/ijerph18020597_

Round 1

Reviewer 1 Report

The work by Park et al. investigated the removal of sulfur dioxide from the simulated gas mixture with the use of a polypropylene contactor in which the gas can react with the alkaline solution media. Overall, the conducted experiments and obtained results are clear, but there is a lack of novelty in this work. Materials that are used are all commercially available, the chemistry of sulfur dioxide removal is simple, the result is as expected without any surprise and new significant findings. Please find some specific comments below:

  1. The whole premise of using the obtained ammonium sulfate as fertilizer is quite unrealistic. Currently, most of the sulfur dioxide produced on numerous power plants is used to produce gypsum (CaSO4) and the amount of its production is huge (hundreds of million metric tons every year). In contrast, yearly production of ammonium sulfate for the needs of fertilization is much smaller (few million metric tons) and there is no need to make hundreds MMt. I recommend to authors to investigate these material needs in the literature and provide the specific numbers, to justify their idea.
  2. Lines 33-34 state that IEA projects the increase of coal usage, but according to the latest data it is not so https://www.iea.org/data-and-statistics/charts/global-coal-demand-by-scenario-2010-2040. Please update your information both about coal and oil as well.
  3. Three parameters for the module used in the work are mentioned twice in Table 1.
  4. The advantage of the contactor is not completely clear. It seems that in order to remove SO2 from the flue gas it would be sufficient to simply bubble the gas through the alkaline liquid, and perhaps more economical because one does not need to push the solution through the contactor. It would be great if you could provide some experimental proof that the utilization of the contactor is indeed necessary.
  5. The purpose of the eq. 1 is not clear as it is not used anywhere in the following calculations. Moreover, sigma is mentioned in the description as thickness but is not included in the equation itself.
  6. Lines 141-143. Please add the information about the real costs here for various materials based on references along with comparative data currently provided.
  7. As you have utilized different adsorbents could you please provide the chemical reactions taking place between SO2 gas and analyte in the liquid phase? Although seem straightforward for chemists but would be useful for readers from other areas.
  8. Please provide explicitly the composition of the simulated seawater used in this work as it is not trivial knowledge.
  9. Two sentences in lines 168-171 repeat the same message.
  10. Lines 203-206. As for me, this seems like an anomaly. If the authors' idea about the corrosion is correct, why would it happen so suddenly and not gradually? Have you tried to repeat this experiment? If so, did you observed the same behavior repeatably?
  11. In eq.2, Qg value is not explained
  12. Line 216 – Please elaborate on what does it mean that pressure can be neglected? It is equal to 0?
  13. Lines 269-271. Although the formation of ammonium sulfate is clear and there is no doubt in that - the practicality of this approach to the formation of fertilizer is doubtful as the concentrations of the solution are very small to be considered as fertilizer. These would require huge amounts of water to be transported with a small amount of actual fertilizer. Please reconsider these statements.
  14. What is the composition of the aqueous mixture at the moment when the adsorbent is changed? According to Fig. 7, it is not pure NH4SO3 - as the reaction does not go to completion. Is it fine to still have ammonia in the solution from the point of view of the toxicity of gaseous NH3?

Author Response

  1. The whole premise of using the obtained ammonium sulfate as fertilizer is quite unrealistic. Currently, most of the sulfur dioxide produced on numerous power plants is used to produce gypsum (CaSO4) and the amount of its production is huge (hundreds of million metric tons every year). In contrast, yearly production of ammonium sulfate for the needs of fertilization is much smaller (few million metric tons) and there is no need to make hundreds MMt. I recommend to authors to investigate these material needs in the literature and provide the specific numbers, to justify their idea.

=> The technology to produce by-product gypsum is a proven technology. However, in the process of producing by-product gypsum, a large amount of carbon dioxide is generated. Besides, energy consumption occurs during the drying process of gypsum dehydration, and eventually, carbon dioxide is generated. There is an issue of depletion of high-quality limestone in Korea, such as CaO and CaCO3, required to produce by-product gypsum. (Lines 52-56)

There is still a need for further consideration as to whether ammonium sulfate prepared in this way can be used as fertilizer, but in this paper we would like to propose this method as an alternative.

  1. Lines 33-34 state that IEA projects the increase of coal usage, but according to the latest data it is not so https://www.iea.org/data-and-statistics/charts/global-coal-demand-by-scenario-2010-2040. Please update your information both about coal and oil as well.

=> We modified it by referring to the latest report in the link you sent us. (Lines 34-36)

  1. Three parameters for the module used in the work are mentioned twice in Table 1.

=> The repeated contents of Table 1 were deleted.

  1. The advantage of the contactor is not completely clear. It seems that in order to remove SO2 from the flue gas it would be sufficient to simply bubble the gas through the alkaline liquid, and perhaps more economical because one does not need to push the solution through the contactor. It would be great if you could provide some experimental proof that the utilization of the contactor is indeed necessary.

=> The advantages of replacing conventional scrubbers with membrane contactors are described in the manuscript, referring to the scientific/technical report of the U.S. GTI Company. (Lines 62-69)

  1. The purpose of the eq. 1 is not clear as it is not used anywhere in the following calculations. Moreover, sigma is mentioned in the description as thickness but is not included in the equation itself.

=> Eq. 1 was inserted to help understand the mass transfer phenomena of gas and absorbent in the membrane contactor but was deleted as it seemed to be irrelevant to the content of this study.

  1. Lines 141-143. Please add the information about the real costs here for various materials based on references along with comparative data currently provided.

=> The price (USD) per meter of hollow fiber membrane according to each material was inserted based on the reference. (Lines 182-186)

  1. As you have utilized different adsorbents could you please provide the chemical reactions taking place between SO2 gas and analyte in the liquid phase? Although seem straightforward for chemists but would be useful for readers from other areas.

=> We inserted the chemical equations for the SO2 removal reaction of various absorbents used in the manuscript with references. (Lines 176-178)

SO2 (g) ↔ SO2 (aq)                                                                         (1)

H2O (aq) + SO2 (aq) ↔ H2SO3 (aq)                                                      (2)

H2O2 (aq) + SO2 (aq) ↔ H2SO4 (aq)                                                    (3)

2NaOH (aq) + SO2 (g) → Na2SO3 (aq) + H2O (aq)                                  (4)

2NaHCO3 (aq) + SO2 (aq) → Na2SO3 (aq) + 2CO2 (g) +H2O (aq)                 (5)

Na2CO3 (s) + SO2 (aq) → Na2SO3 (aq) + CO2 (g)                                     (6)

Na2SO3 (s) + SO2 (aq) + H2O (aq) → 2NaHSO3 (aq)                                (7)

2NH4OH (aq) + SO2 (aq) + H2O (aq) ↔ (NH4)2SO3 (aq) + 2H2O (aq)            (8)

  1. Please provide explicitly the composition of the simulated seawater used in this work as it is not trivial knowledge.

=> In lines 113-115, the information on the sea-salt used in this study has been inserted.

  1. Two sentences in lines 168-171 repeat the same message.

=> We deleted unnecessary, repeated messages.

  1. Lines 203-206. As for me, this seems like an anomaly. If the authors' idea about the corrosion is correct, why would it happen so suddenly and not gradually? Have you tried to repeat this experiment? If so, did you observed the same behavior repeatably?

=> The corrosion of the membrane contactor potting caused by ammonium carbonate did not occur suddenly, but rather slowly. The corrosion of the polyurethane gradually proceeded with each replacement cycle of the absorbent. An overflow of gas was observed in the potting of the membrane contactor after 100 hours of operation. The corrosion phenomenon of potting of the membrane contactor results from confirmation through repeated experiments and data on the chemical stability of polyurethane. This content has been inserted into the manuscript along with a description of the potting materials and chemical stability of the membrane contactor. (Lines 249-256)

  1. In eq.2, Qg value is not explained

=> We inserted a description of the Qg value. (Line 271)

  1. Line 216 – Please elaborate on what does it mean that pressure can be neglected? It is equal to 0?

=> We revised the manuscript as follows. ‘The membrane contactors do not depend on the pressure during operation. The force applied to the membrane contactor is 1 atm, and the pressure difference is close to zero’ (Lines 273-276)

  1. Lines 269-271. Although the formation of ammonium sulfate is clear and there is no doubt in that - the practicality of this approach to the formation of fertilizer is doubtful as the concentrations of the solution are very small to be considered as fertilizer. These would require huge amounts of water to be transported with a small amount of actual fertilizer. Please reconsider these statements.

=> We agree that a small amount of ammonium sulfate fertilizer is produced. However, using ammonium sulfate in powder as fertilizer causes a decrease in the resistance to pests and cold hazards due to excess nitrogen. Also, it interferes with the growth of living things and reduces yields. To prevent these problems and denitrification, they should be dissolved in water and used several times. (Lines 331-335) Besides, since the resulting aqueous ammonium sulfate solution contains a large amount of moisture, it is expected that the problem of transportation can be solved by installing a pipe and supplying it to nearby farms. (Added content to Lines 338-340)

  1. What is the composition of the aqueous mixture at the moment when the adsorbent is changed? According to Fig. 7, it is not pure NH4SO3 - as the reaction does not go to completion. Is it fine to still have ammonia in the solution from the point of view of the toxicity of gaseous NH3?

=> When the absorbent was replaced, the aqueous mixture contains ammonium carbonate ((NH4)2CO3), which absorbed a small amount of carbon dioxide. Ammonium carbonate is spontaneously decomposed into ammonium bicarbonate (NH4HCO3) and ammonia (NH3). The decomposed NH3 reacts with water to exist in the form of NH4OH. (Lines 245-248)

 The reaction equation is as follows:

NH4OH + CO2 + H2O → (NH4)2CO3

(NH4)2CO3 → NH4HCO3 + NH3

NH3 + H2O → NH4OH

Ammonia is easily soluble in water, so there is no need to consider gaseous ammonia toxicity.

Reviewer 2 Report

This study used PP membrane reactor and ammonia solution to remove SO2 and CO2 from flue gas. The topic fits the journal scope, and the article is well organized. I think the article could be published in this journal if the following points are properly addressed.

The term ammonia water is not professional. Please rephrase it.

It would be necessary to state the temperature of flue gas, and discuss if the PP membrane is able to stand this temperature. If not, is cooling a required step?

The authors used PP membrane, is it porous? What is the pore size distribution? This question arises from Fig. 2. Because Fig. 2 showed the membrane is porous. From my experience, PP membrane is generally regarded as dense membrane.

Author Response

This study used PP membrane reactor and ammonia solution to remove SO2 and CO2 from flue gas. The topic fits the journal scope, and the article is well organized. I think the article could be published in this journal if the following points are properly addressed.

  1. The term ammonia water is not professional. Please rephrase it.

=> The term has been revised to 'ammonia solution'.

  1. It would be necessary to state the temperature of flue gas, and discuss if the PP membrane is able to stand this temperature. If not, is cooling a required step?

=> The flue gas temperature is 90-140℃, and the operating temperature of the membrane contactors used is below 80℃ or less. When real flue gas is used, it is planned to replace it with membrane contactors applied to high temperatures, or operating temperatures will be adjusted through heat exchangers. (Lines 349-352)

  1. The authors used PP membrane, is it porous? What is the pore size distribution? This question arises from Fig. 2. Because Fig. 2 showed the membrane is porous. From my experience, PP membrane is generally regarded as dense membrane.

=> The membrane contactor used in this study was a porous hollow fiber membrane. A detailed description of the membrane contactor, including pore size and porosity, was inserted in lines 134-141 based on the reference.

Reviewer 3 Report

The manuscript present an interesting concept of using PP hollow fiber membrane contactor as the contact material in removing sulphur dioxide with various absorbents. However, the presentation of result unable clear assessment which operating conditions presented in the manuscript really influence the SO2 removal. Therefore, the final decision about possible publication can be made after major improvement which would include the following major points.

Major points:

  1. Section 3.2. and Fig. 3. It would be interesting to see how the sulphur concentration in absorbent changed over the time, and how far it is from maximum possible concentration of sulphur in each absorbent. Therefore, it would be possible to assess whether the maximum capacity for sulphur for each absorbent was achieved.
  2. Section 3.3 and Fig. 4. Since the liquid was recycled it is important to show explicitly the number of turnovers of liquid and the final concentration of sulphur in absorbent. The change in liquid flow rate can influence the typical membrane’s mass transfer phenomena like concentration polarization or fouling, which ultimately affects the overall process performance. However before assessing that, it would be necessary to see the change in sulphur concentration in absorbent over the processing time.
  3. Lines 207-211. What is the point of using such material if it is not stable? I do appreciate that authors mentioned that but please provide more reasoning about it, or show explicitly that you discovered/noticed that during your experiments. Bad results are still good results in the sense of presenting them and aware everybody else.
  4. Lines 246-247. How the power consumption was assessed/calculated? Please provide more details in the text.
  5. Lines 263-271. In reality, when real flue gas would be used, some hazardous substances (like heavy metals, dioxins, etc.) could be also absorbed therefore that remark should be carefully included.

Minor points

  1. Line 65. Is NaOH a salt?
  2. Please check style and English. For example, lines 84-87.
  3. 1. Check multiplication sign.
  4. Line 274. It would be better to highlight that authors used a “model of flue exhaust gas” instead of “flue exhaust gas”.

Author Response

The manuscript present an interesting concept of using PP hollow fiber membrane contactor as the contact material in removing sulphur dioxide with various absorbents. However, the presentation of result unable clear assessment which operating conditions presented in the manuscript really influence the SO2 removal. Therefore, the final decision about possible publication can be made after major improvement which would include the following major points.

* Major points:

  1. Section 3.2. and Fig. 3. It would be interesting to see how the sulphur concentration in absorbent changed over the time, and how far it is from maximum possible concentration of sulphur in each absorbent. Therefore, it would be possible to assess whether the maximum capacity for sulphur for each absorbent was achieved.

=> The change in sulfur concentration in the absorbent over time is significant for evaluating absorbent saturation (In Figure 3, a graph of the sulfur concentration of the absorbent overtime was inserted.). However, when selecting the SO2 absorbent in this paper, the key was how well the high SO2 removal efficiency was maintained over time.

‘The same trend can be confirmed by looking at the SO2 loading of the absorbent over time in Figure 3.’ (Inserted in manuscript, Lines 182-183)

  1. Section 3.3 and Fig. 4. Since the liquid was recycled it is important to show explicitly the number of turnovers of liquid and the final concentration of sulphur in absorbent. The change in liquid flow rate can influence the typical membrane’s mass transfer phenomena like concentration polarization or fouling, which ultimately affects the overall process performance. However before assessing that, it would be necessary to see the change in sulphur concentration in absorbent over the processing time.

=> In the experimental results shown in Figure 4, the absorbent was not reused. These are the results of individual experiments under each condition. (Corrected the sentence, lines 204-207)

 As stated in #1, we added the SO2 concentration plot to Figure 3. However, the max saturation sulfur concentration cannot be confirmed through this experiment. As discussed in Figure 3, the same trend will be produced, so the sulfur concentration in absorbent is not indicated separately.

  1. Lines 207-211. What is the point of using such material if it is not stable? I do appreciate that authors mentioned that but please provide more reasoning about it, or show explicitly that you discovered/noticed that during your experiments. Bad results are still good results in the sense of presenting them and aware everybody else.

=> Additional information on the stability to CO2 for the membrane contactors potting material has been inserted in lines 245-260 based on the references.

  1. Lines 246-247. How the power consumption was assessed/calculated? Please provide more details in the text.

=> Comparing the energy consumption of the conventional packed column and the membrane contactor system was modified based on the reference. (Lines 307-311)

  1. Lines 263-271. In reality, when real flue gas would be used, some hazardous substances (like heavy metals, dioxins, etc.) could be also absorbed therefore that remark should be carefully included.

=> In the case of real flue gas applications, a description of the pretreatment facility for minimizing pollutants has been inserted in lines 343-352 based on references.

* Minor points

  1. Line 65. Is NaOH a salt?

=> Corrected the sentence. (Deleted ‘salt’)

  1. Please check style and English. For example, lines 84-87.

=> We once again checked the style and English of the manuscript.

  1. Check multiplication sign.

=> We modified it according to the advice.

  1. Line 274. It would be better to highlight that authors used a “model of flue exhaust gas” instead of “flue exhaust gas”.

=> We modified it according to the advice.

Round 2

Reviewer 1 Report

The authors have improved the manuscript. I have just a few comments related to the previous questions and replies. They are given as italic text following the authors replies.

  1. Lines 269-271. Although the formation of ammonium sulfate is clear and there is no doubt in that - the practicality of this approach to the formation of fertilizer is doubtful as the concentrations of the solution are very small to be considered as fertilizer. These would require huge amounts of water to be transported with a small amount of actual fertilizer. Please reconsider these statements.

=> We agree that a small amount of ammonium sulfate fertilizer is produced. However, using ammonium sulfate in powder as fertilizer causes a decrease in the resistance to pests and cold hazards due to excess nitrogen. Also, it interferes with the growth of living things and reduces yields. To prevent these problems and denitrification, they should be dissolved in water and used several times. (Lines 331-335) Besides, since the resulting aqueous ammonium sulfate solution contains a large amount of moisture, it is expected that the problem of transportation can be solved by installing a pipe and supplying it to nearby farms. (Added content to Lines 338-340)

I do not argue that using powder fertilizer has some negative consequences. However, it is important to indicate - what maximum concentration in solution could be achieved. Also, transportation via the pipeline to nearby farms is unlikely, firstly it seems that even one conventional coal power plant produces a very large amount of fertilizer, and secondly, most of the farms are not located in the vicinity of coal power plants. For instance in the area I live there are no coal power plants, electricity is made by nuclear, but there are lots of farms that need fertilizer. What option would you suggest for them? I believe you have to consider the production of dry fertilizer from the solution just for the sake of transportation (dissolution can be performed on-site). 

  1. What is the composition of the aqueous mixture at the moment when the adsorbent is changed? According to Fig. 7, it is not pure NH4SO3 - as the reaction does not go to completion. Is it fine to still have ammonia in the solution from the point of view of the toxicity of gaseous NH3?

=> When the absorbent was replaced, the aqueous mixture contains ammonium carbonate ((NH4)2CO3), which absorbed a small amount of carbon dioxide. Ammonium carbonate is spontaneously decomposed into ammonium bicarbonate (NH4HCO3) and ammonia (NH3). The decomposed NH3 reacts with water to exist in the form of NH4OH. (Lines 245-248)

Okay, I understood that ammonia will stay dissolved in water, but don't you need to remove it before the utilization of fertilizer?  

Author Response

The authors have improved the manuscript. I have just a few comments related to the previous questions and replies. They are given as italic text following the authors replies.

  1. Lines 269-271. Although the formation of ammonium sulfate is clear and there is no doubt in that - the practicality of this approach to the formation of fertilizer is doubtful as the concentrations of the solution are very small to be considered as fertilizer. These would require huge amounts of water to be transported with a small amount of actual fertilizer. Please reconsider these statements.

=> We agree that a small amount of ammonium sulfate fertilizer is produced. However, using ammonium sulfate in powder as fertilizer causes a decrease in the resistance to pests and cold hazards due to excess nitrogen. Also, it interferes with the growth of living things and reduces yields. To prevent these problems and denitrification, they should be dissolved in water and used several times. (Lines 331-335) Besides, since the resulting aqueous ammonium sulfate solution contains a large amount of moisture, it is expected that the problem of transportation can be solved by installing a pipe and supplying it to nearby farms. (Added content to Lines 338-340)

I do not argue that using powder fertilizer has some negative consequences. However, it is important to indicate - what maximum concentration in solution could be achieved. Also, transportation via the pipeline to nearby farms is unlikely, firstly it seems that even one conventional coal power plant produces a very large amount of fertilizer, and secondly, most of the farms are not located in the vicinity of coal power plants. For instance in the area I live there are no coal power plants, electricity is made by nuclear, but there are lots of farms that need fertilizer. What option would you suggest for them? I believe you have to consider the production of dry fertilizer from the solution just for the sake of transportation (dissolution can be performed on-site). 

=> Ammonium sulfate fertilizer is generally suitable for crops such as common wheat, corn, rice, cotton, sweet potato, sesame seeds, fruit trees, and vegetables. The appropriate amount of fertilizer used depends on the crop. In the case of wheat, 62 kg of ammonium sulfate fertilizer per 0.1 ha is used. It is considered that a sufficient supply of ammonium sulfate aqueous solution is possible by increasing the concentration of the absorbent used in this study or by scale-up the reaction facility. (Lines 310-315)

Nitrogen

Ammonium sulfate (kg/0.1ha)

Wheat

12-14

62

Corn

18

86

Rice

11-16

55-74

Sweet potato

6-7

31

Sesame

4-5

21

Vegetables

16-32

86-150

Fruit trees

13-25

67-119

As pointed out, the supply of an aqueous ammonium sulfate fertilizer through pipeline installation at nearby farms is limited. Therefore, if applied in practice, an aqueous ammonium sulfate solution should be transported to the tank lorry for use, or additional facilities (e.g., dryer, crystalizer) should be used in the production process to supply it in powder form (Lines 321-325).

  1. What is the composition of the aqueous mixture at the moment when the adsorbent is changed? According to Fig. 7, it is not pure NH4SO3 - as the reaction does not go to completion. Is it fine to still have ammonia in the solution from the point of view of the toxicity of gaseous NH3?

=> When the absorbent was replaced, the aqueous mixture contains ammonium carbonate ((NH4)2CO3), which absorbed a small amount of carbon dioxide. Ammonium carbonate is spontaneously decomposed into ammonium bicarbonate (NH4HCO3) and ammonia (NH3). The decomposed NH3 reacts with water to exist in the form of NH4OH. (Lines 245-248)

Okay, I understood that ammonia will stay dissolved in water, but don't you need to remove it before the utilization of fertilizer?  

=> Yes. There is no need to remove it. Ammonia nitrogen means nitrogen in the form of NH4 or NH3 ions, and ammonia dissolved in water also exists as ammonia nitrogen used as fertilizer, so there is no need to remove it. However, it should be used following the appropriate concentration for each crop, and it is not suitable for acidic soils, so it should be used only in neutral or alkaline soils. Also, when mixed with basic fertilizer, ammonia volatilizes, so it should not be mixed. This content is inserted in section 3.6 at lines 303-308.
